# Merging transformation optics with electron-driven photon sources

Nahid Talebi [1], Sophie Meuret[2], Surong Guo[1], Mario Hentschel[3], Albert Polman[2], Harald Giessen[3] & Peter A. van Aken[1]

Relativistic electron beams create optical radiation when interacting with tailored nanostructures. This phenomenon has been so far used to design grating-based and holographic electron-driven photon sources. It has been proposed recently that such sources can be used for hybrid electron- and light-based spectroscopy techniques. However, this demands the design of a thin-film source suitable for electron-microscopy applications. Here, we present a mesoscopic structure composed of an array of nanoscale holes in a gold film which is designed using transformation optics and delivers ultrashort chirped electromagnetic wave packets upon 30–200 keV electron irradiation. The femtosecond photon bunches result from coherent scattering of surface plasmon polaritons with hyperbolic dispersion. They decay by radiation in a broad spectral band which is focused into a 1.5 micrometer beam waist. The focusing ability and broadband nature of this photon source will initiate applications in ultrafast spectral interferometry techniques.

[1] Stuttgart Center for Electron Microscopy, Max Planck Institute for Solid State Research, Heisenbergstr. 1, Stuttgart 70569, Germany. [2] Center for Nanophotonics, AMOLF, Science Park 104, Amsterdam 1098 XG, The Netherlands. [3] 4th Physics Institute and Research Center SCoPE, University of Stuttgart, Pfaffenwaldring 57, Stuttgart 70569, Germany. Correspondence and requests for materials should be addressed to N.T. (email: n.talebi@fkf.mpg.de)

The ability to control light at the nanoscale with precisely engineered nanostructures has triggered various applications in quantum optics[1,2], nano-circuitry[3–5], and microscopy[6–8]. In particular, realization of a small focus spot in air beyond the reach of evanescent waves is most appealing, and this can be achieved by Fresnel lenses[9], planar metallic metasurfaces[10,11], and photon sieves[12,13], with all of them acting as diffractive lenses. Additionally, non-diffractive lenses, controlling the effective response of the medium, have also been proposed and realized using the principles of transformation optics[14,15]. This gives great ease in controlling the direction and polarization states of the light[16,17]. However, most of the incorporated structures may be used to control plane-wave light, while their application to control radiation from single nano-emitters is not straightforward. Therefore, in order to control the polarization, intensity, and direction of light emitted from localized radiation sources and directed electron beams in free space, different design strategies are required.

Radiation mechanisms of electron beams interacting with either magnetic undulators or materials are numerous, including Larmor radiation[18], transition[19] and plasmon-induced radiation[20], Smith–Purcell radiation[21–24], and Bremsstrahlung[25]. Transition radiation occurs when an electron traverses a metallic surface, due to the sudden annihilation of the induced dipole formed by the electron and its image charge. In addition to transition radiation, electrons launch plasmon polaritons along the surface of a metal. These induced polaritons can couple to radiation by scattering from gratings and defects. Massive ultrabroadband radiation sources such as free-electron lasers are based on ultra-relativistic electron beams on the order of GeV[26,27]. Differently, slower electrons interacting with optical and plasmonic gratings will emit coherent Smith–Purcell radiation which can interfere with the plasmon-induced radiation[28]. Such a mechanism of radiation might be also used to realize a miniaturized electron-driven photon source (EDPHS), though the large angular distribution of the emitted photons from the interaction of nonrelativistic electrons with nanostructures imposes the incorporation of feedback elements or closed waveguide geometries[29–31]. Recently, diffractive metamaterial lenses have been also applied to control the directionality and the polarization states of electron-induced radiation[32–34]. Two-dimensional arrays of split-ring resonators have been utilized for enhancing the radiation at a rather wide spectral range (0.6 eV bandwidth) due to the broad plasmonic resonances in gold. However, the emission pattern of the EDPHS based on these resonators sustains a wide angular range. In contrast, holographic designs are perfectly suited for controlling the directionality of the emission[33]. In this approach, the interference pattern of the electron-induced plasmons at the gold/air interface with a light field at a specific wavelength and with a desired shape is used to generate the required hologram; however, this method is highly frequency-selective.

In this work, we propose a microscale design principle for controlling the directionality of electron-induced radiation. We avoid diffractive metamaterial elements[18,19] in order to achieve ultrafast control and unidirectional emission and exploit hyperbolic dispersion to create efficient plasmon radiation. Our design principle is rather based on the geometrical manifestation of the principles of transformation optics[14], by manipulating the effective refractive index of the thin film using incorporated holes with dimensions much smaller than the wavelength of the emitted light pulse. Additionally, the realized system mimics an anisotropic thin film, with different signs for in-plane and normal permittivity components, thus the structure is effectively a hyperbolic material. Consequently, a branch of polaritons propagating along the surface of our engineered thin film is mapped into the light cone, which largely enhances the electron-induced

radiation. We thus achieve focused electron-induced radiation using ultrathin planar geometries within a broad spectral range centered around 0.6 eV covering a bandwidth of 0.8 eV (Fig. 1a). Electron energy-loss spectroscopy and angle-resolved cathodoluminescence have been used to characterize the generated light fields and are in good agreement with finite-difference time-domain simulations. Our design principle facilitates the direct incorporation of the introduced EDPHS inside an electron microscope, to further pave the way for concomitant electron–photon spectroscopy and interferometry[35].

## Results

**Design principles**. Previously, it was shown that a swift electron interacting with a superlens, composed of a hexagonal two-dimensional photonic crystal incorporated into an inverted hemispherical geometry, will emit focused transition radiation[35]. A simpler planar structure, which in an effective medium approximation resembles the same behavior, can be realized by projecting the location of the photonic crystal elements from the curved spherical geometry into the desired plane, which is located directly above the hemisphere (Fig. 1b). Mathematically, the transformation from the spherical plane to the cylindrical plane leads to the modification of the permittivity as $\varepsilon' = |\Lambda|^{-1} \Lambda \varepsilon \Lambda^T$, where $\Lambda$ is the Jacobian of the transformation matrix, and $\varepsilon$ is the initial permittivity of the material in the spherical coordinate system. The inhomogeneous planar refractive index of the plane by projection from the spherical system to the cylindrical coordinate system is obtained as $\varepsilon'_{\rho\rho} = \varepsilon \rho^2 (\rho^2 + d^2)^{-1}$, where $\rho = (x^2 + y^2)^{\frac{1}{2}}$. For $\rho \gg d$, $\varepsilon'_{\rho\rho} = \varepsilon$. We also assume that $\mu = \mu' = 1$, for the simplicity of the fabrication processes. A geometrically discrete coordinate transformation is applied. A hexagonal lattice of point defects (i.e., holes) is mapped from the spherical coordination to the cylindrical system, which leads to the distribution of points as illustrated in the bottom plane in Fig. 1b (gray dots). In order to keep the system cylindrically symmetric, holes are only formed inside a certain ring diameter. Throughout the paper we use the finite-difference time-domain (FDTD) method for calculating the electron-induced radiation[36] (see method section). A snapshot of the z-component of the electron-induced radiation after the interaction of a 30 keV electron with a hemispherical gold film (with dimensions described below) is illustrated in Fig. 1c. A hexagonal lattice of holes with a diameter of 100 nm and rim to rim distance of 50 nm is drilled into the spherical film. The radiation is in the form of a focused ultrafast electromagnetic wave packet with the focal point positioned at the center of the sphere. It originates from the radiation damping of surface plasmon polaritons (SPPs) in our spherical film by the incorporated lattice of holes, in addition to the transition radiation (Supplementary Note 1, Supplementary Fig. 1a, b). While the radiation damping of SPPs in a planar gold film is negligible, for a thin film with the lattice of holes the effective permittivity is anisotropic and as a result the SPP dispersion is mapped into the light cone. Hence radiation damping is observed (Supplementary Fig. 1c and d). This behavior in addition to the spherical curvature of the film in Fig. 1c causes an enhanced radiation which is focused at the center of the sphere.

The calculated radiation spectrum (cathodoluminescence, CL) for a plane located 5 microns below the hemispherical lens is shown in Fig. 1d and extends from 0.4 to 2.6 eV. The spectrum is calculated using the Poynting vector $\Gamma^{CL}(\omega) = (2\hbar\omega)^{-1} \times \text{Re} \int \int_s \mathbf{E}(\mathbf{r}, \omega) \times \mathbf{H}^*(\mathbf{r}, \omega) \cdot \mathbf{ds}$, where $\mathbf{E}$ and $\mathbf{H}$ are the scattered components of the electron-induced electric and magnetic fields, respectively, $\omega$ is the angular frequency of the emitted light, and the plane $s$ is located in the far field. This formulation allows for a direct comparison with electron energy-loss spectroscopy (EELS)

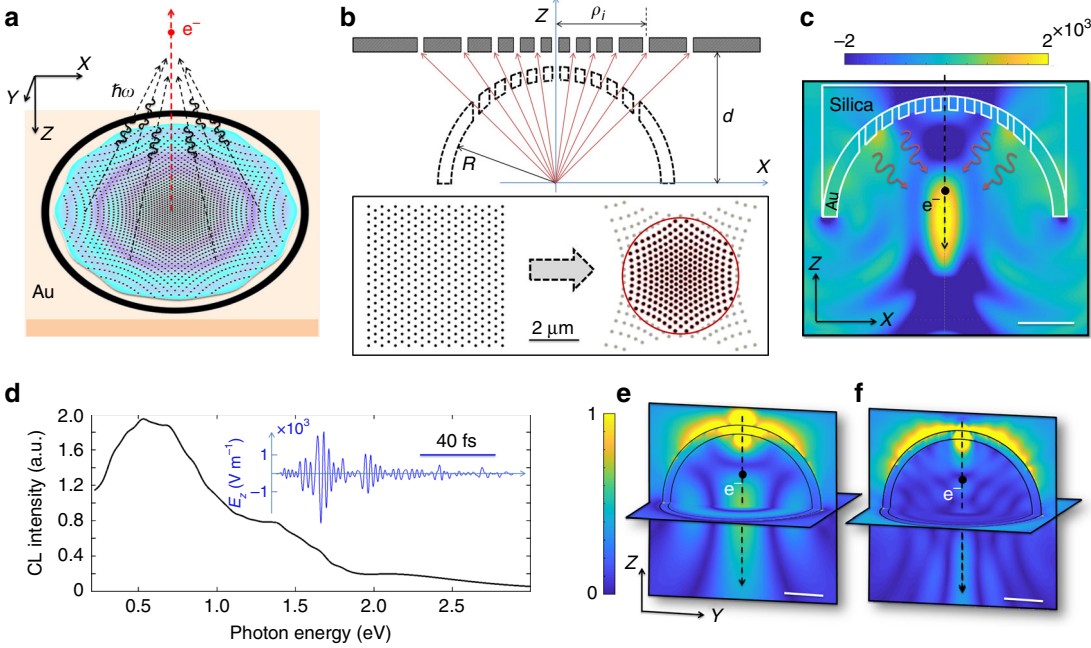

**Fig. 1** Design Principle for the broadband light source. **a** Formation of focused broadband transition radiation in interaction of an electron beam with an engineered planar lens. The structure is designed to effectively mimic a porous hemispherical geometry. **b** Schematic of the design principles that includes projection of incorporated holes in the hemispherical lens to a planar gold film. **c** Interaction of a 30 keV electron (incident from the top) with a mesoscopic hemispherical porous film positioned on a silica substrate. The electron passes through one of the holes at the center. Depicted in the figure is a snapshot of the simulated $z$-component of the electric field 22.55 fs after impact on the lens surface which indicates a clear focus. Color bar is in the dimensions of $Vm^{-1}$ and scale bar is 1 μm. **d** CL spectra of the hemispherical structure computed at a plane positioned 5 micrometer below the structure. Inset: Temporal representation of the $z$-component of the electric field at the focal point in **c**. Time-averaged electron-induced electromagnetic wave packet at the energies **e** $E = 0.6$ eV and **f** $E = 1.3$ eV. Depicted is the magnitude of the Fourier transform of the $z$-component of the electric field in the $xy$ and $yz$ planes. Color bar is in arbitrary units and scale bars are 1 μm

using Poynting's theorem[29]. Thus only the components normal to the plane of the detector (transverse components) are relevant to the CL spectrum[37]. The fact that the emitted light constitutes many frequency components is also reflected by the computed temporal distribution of the $z$-component of the electric field at the focal point (center of the sphere, see the inset in Fig. 1d). The time-averaged $z$-component of the electric field magnitude in the $xy$ and $yz$ planes at the energies of $E = 0.6$ eV and $E = 1.3$ eV are shown in Fig. 1e, indicating focused radiation at the center of the spherical geometry at both photon energies.

**Structure**. Our planar metasurface lens has a spectral response that peaks in the visible-near infrared spectral range (0.5–2.0 eV). We use a hexagonal lattice of holes, each with a diameter of 100 nm and a center-to-center distance of 150 nm, projected onto a planar Au disc with a thickness of 50 nm, using the above mentioned algorithm. The whole structure is placed on a 20 nm thick $Si_3N_4$ membrane. The radius of the original hemisphere is $R = 5.7$ μm, and $d = R$, which leads to a focal length $f = R$ for the planar lens. Scanning Transmission Electron Microscopy (STEM) high-angle annular dark field images of the realized planar lens are shown in Fig. 2a, b. A void ring is incorporated at the circumference of the lens in order to facilitate the reflection of the propagating SPPs and hence to increase the generation of far-field radiation, as shown below. The inserted void ring has a strong effect on the photon emission capability of the lens, which is understood by comparing theoretically and experimentally the response of structures by including and excluding the ring (see Supplementary Note 2 and Supplementary Fig. 2).

**Electron energy-loss spectroscopy investigations**. We first investigate the near-field behavior of the fabricated planar lens using electron energy-loss spectroscopy (EELS) at the unique Zeiss SESAM transmission electron microscope[38] in the STEM mode. In EELS, due to the inelastic interaction of the electron beam with the optical modes of the nanostructure, the electron will lose energy, where the amount of the energy-loss is measured by using an energy-loss spectrometer. The EELS signal reflects the probability of the electron beam to excite optical modes during its interaction with the sample and thus senses the photonic local density of states projected onto the trajectory of the electron from a few tens of meV to hundreds of eV[39]. For this reason, EELS is ideally suited for probing ultra-broadband excitations. The near-field resonances should also exhibit similarly broad features as the far-field radiation. The measured and simulated EELS spectra, taken at different locations on the lens, are shown in Fig. 2c, and show a broad spectral feature at a central energy of $E = 0.6$ eV and a bandwidth of $\delta E = 0.8$ eV. An asymmetric resonance is observed due to the high-order chirping of the electron-induced polarizations. The chirping effect is understood by a gradual change in the frequency of the light-field oscillations, as will be described later. Interestingly, the EELS spectra at positions closer to the rim of the structure exhibit broader features than near the center. However, the highest intensity for the EELS signal occurs at the center of holes, as shown in the Supplementary Fig. 3 (See Supplementary Note 3 for further explanations).

The electron energy-loss is calculated using an overlap integral (between the current distribution of the electron beam and the scattered electric field projected along the trajectory of the electron) which within the non-recoil approximation is simplified as $\Gamma^{EELS}(x_0, y_0, \omega) = e(\pi\hbar\omega)^{-1} \times \mathrm{Re}\tilde{E}_z(x_0, y_0, k_z = \omega v_e^{-1}; \omega)$, where

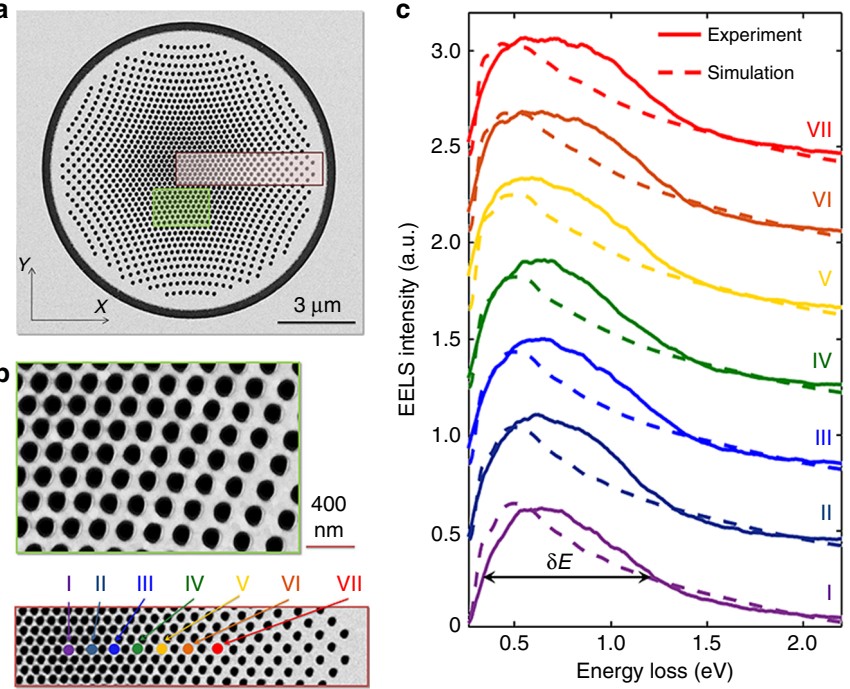

**Fig. 2** Characterization of the fabricated planar lens. **a** Dark field STEM image of the fabricated lens. **b** The same image at higher magnification from the green and red boxes in **a**. **c** Experimental and simulated EELS spectra (200 keV primary electron beam energy) at the positions depicted by colored spots indicated in **b**

$(x_0,y_0)$ is the impact location of the electron beam in the transverse direction, $e$ is the elementary charge, $k_z$ is the projected wavenumber of the scattered field along the electron trajectory, $\tilde{E}_z$ is the Fourier transform of the electric field in both spectral and spatio-spectral domains, and $v_e$ is the velocity of the electron[40]. The simulated EELS spectra, as shown in Fig. 2c with dashed lines, show good agreement with the measured spectra. However, experimental EELS data show slightly broader resonances, which are due to the convolution of the zero-loss peak with plasmon resonances[41]. This zero-loss peak has contribution from electrons that did not undergo inelastic scattering, but may have been scattered elastically or with an energy loss too small to measure. Moreover, surface roughness in the sputtered gold thin film increases the damping of surface plasmons and hence can result in an additional broadening of the EELS spectra.

**Radiation mechanism**. To obtain further insight into the radiation mechanism of the planar lens interacting with the moving electron we investigate the temporal dependence of the field evolution (Fig. 3a and Supplementary Movie 1). We use 30 keV electrons, as in CL measurements described further below. The electron excites the planar lens when it reaches the near-field region of the structure. The interaction of the electron with the thin film is taking place only within a sub-fs time scale; however this interaction induces relatively long lasting oscillations in the sample with a relaxation time of $\tau \approx 20$ fs. The contribution of the induced polarization to the radiation is not gradual in time. The first ultrafast pulse, due to transition radiation, leaves the structure directly upon electron impact and in the form of a sub-cycle spherical wave front, overtaking the electron at only a sub-fs time scale (Fig. 3b)[20]. It should be noticed, however, that the duration of the transition radiation is very much dependent on the thickness of the film (two dipoles created and annihilated at upper and lower surfaces) and the presence of substrate. Additionally, the amplitude and position of the image charge is also altered by the presence of the holes.

Simultaneously, the electron excites propagating SPPs of tailored dispersion due to the inhomogeneous lattice of holes. The excited SPPs gradually scatter out due to their hyperbolic dispersion, with the $z$-component coupling to the far-field, thus build up the continuum of radiation modes. Due to the separation in time the transition and SPP radiations will not interfere for our structure, dissimilar to the observations for grating-facilitated out-coupling of SPPs[42,43]. Interestingly, the lower frequency components leave the lens sooner, followed by a gradual chirping of the entire radiation towards higher frequency components (see Fig. 3b). The time lag between the transition radiation and the plasmon-induced radiation in the present case is $\Delta t = 15.6$ fs. The spatial representation of the field components in the frequency domain, however, demonstrates this fact even better (see Fig. 3c). Similar to the case of the hemispherical resonator, our planar lens exhibits a focal waist in the frequency range between 0.6 eV and 1.4 eV (Fig. 3c). Moreover, the electron-induced excitation in forward direction comprises about 43% of the entire electromagnetic energy delivered into the system, where 37% of the electromagnetic energy is dissipated inside the gold film, and 20% is converted into the radiation in the upward direction (data not shown). Interestingly, the radiation spectra and field distributions in the forward and backward directions are not symmetric, despite the symmetry of the structure. This fact is mainly due to the Doppler effect imposed by the moving electron source. The calculated time-averaged amplitude of the electric field at the focal plane is shown in Fig. 4, left column. The $\mathbf{k}$−space distribution is obtained by mapping the field components from the space-time ($\mathbf{r},t$) to the ($\mathbf{k},\omega$) domain, using a spatial Fourier transformation. However, the fact that an FDTD simulation is performed in a closed domain imposes some restrictions over the integration range of the Fourier transformation. To minimize these errors, we multiplied the field components in the ($\mathbf{r},t$) domain with a cylindrically symmetric step window $\Theta(\rho/a)$, where $a$ is the radius of the cylindrical window which was estimated by an ad-hoc procedure, to obtain smooth functions

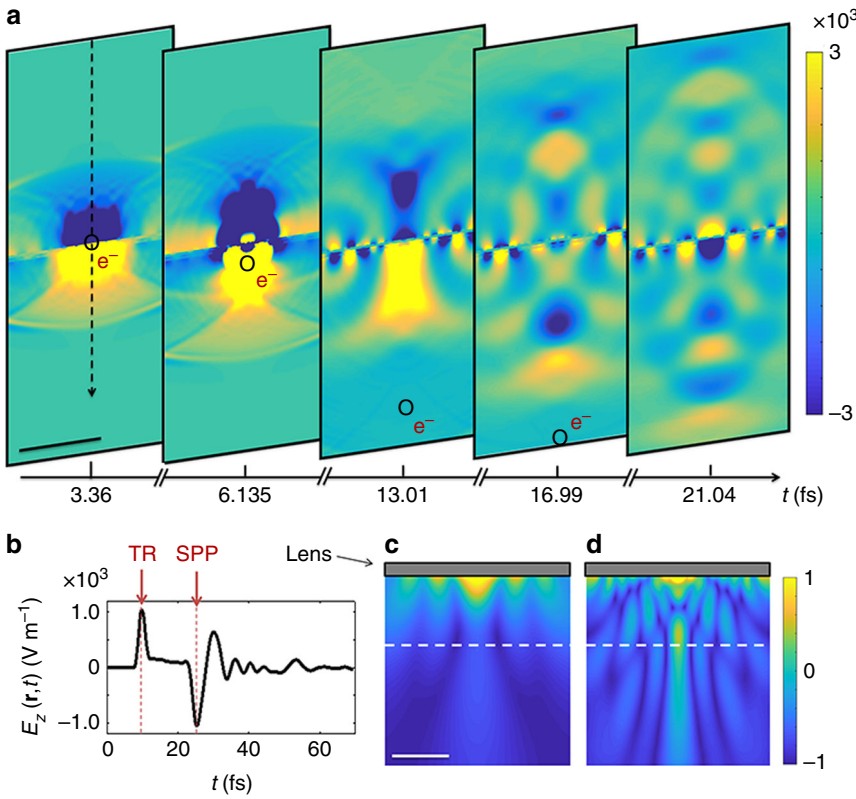

**Fig. 3** Simulated spatio-temporal response of the planar lens interacting with a 30 keV electron. **a** Snapshots of the z-component of the electric field amplitude at depicted times for the indicated electron positions, which demonstrate that the electromagnetic wave packet is focused into a 1.5 μm waist at 5.7 μm distance from the structure (see Supplementary Movie 1). Color bar is the electric field amplitude at the dimensions of V m$^{-1}$. Scale bar is 5 μm. **b** Temporal representation of the z-component of the electric field at the focal point. Fourier transform of the temporal data, giving the z-component of the electric field at the **c** E = 0.6 eV and **d** E = 1.4 eV. The white dashed line marks the focal plane. TR stands for transition radiation. Color bar is the averaged electric field magnitude at arbitrary units. Scale bar is 4 μm

with the best signal-to-noise ratio in the $(\mathbf{k}, \omega)$ domain, but also to maintain all the important spatial features in the space-time domain. The amplitude of the electric field as well as the total electromagnetic energy (Fig. 4 right column) show a high intensity signal at the center of momentum space at the focal plane. The size of this focus spot is largest at E = 0.4 eV; it drastically decreases when increasing frequency. The maximum of the total electromagnetic energy remains located at the center of reciprocal space at photon energies in the range 0.2 eV–1.6 eV. However, the transverse components of the electromagnetic fields that determine the Poynting vector and hence the CL angular profile for the radiation to the far field exhibits opposite behavior (see Fig. 4, right column): high intensity is observed for large angle, in particular for the higher energies. This is to be expected, as longitudinal components of a field always demonstrate a different symmetry than transverse components. For example, for transverse magnetic fields (for the TM$_z$ modes) in a cylindrical coordinate system, $E_\rho \propto \partial^2 E_z/\partial\rho\,\partial z$ and $E_\varphi \propto \partial^2 E_z/\rho\,\partial\varphi\,\partial z$. In other words, the $\phi$−component of the electric field should vanish, as the symmetry of the excitation and the lens requires $\partial/\partial\phi = 0$. $E_\rho$ is, however, present and exhibits different symmetry to $E_z$, which explains the vanishing central bright spot for the CL map for high energies. For computing the transverse components of the electron-induced radiation at the far-field region, the effective current distributions have been calculated along a fictitious plane 50 nm below the sample using the electromagnetic equivalence principle[44]. Those current distributions have been then used to calculate the radiation at the far-field.

**Cathodoluminescence investigations**. We measure the angle-resolved cathodoluminescence emission of the lens using a SEM equipped with a Schottky field-emission electron source operated at 30 keV. The emitted light is collected using an aluminum paraboloid mirror with a focal distance of 0.5 mm which provides a collection solid angle of 1.46 π sr[34]. The mirror is mounted above the sample, hence collecting the photons emitted in the backward direction as depicted at Fig. 5a. The CL measurements have been performed at a photon energy of E = 1.9 eV (wavelength λ = 650 nm). For calculating the CL map, however, the Poynting vector along the r—direction ($S_r$) has been considered. At this energy, the CL map exhibits 8 centrosymmetric bright rings when the electron traverses the structure closed to the center. Additional measurements (see Supplementary Note 5 and Supplementary Fig. 6) show the number of rings decreases for decreasing energy. This behavior points at a geometrical inter-ference effect, caused by the reflection of the SPPs from the inhibited void ring surrounding the lens. This is also understood by the constructive interference of the optical radiation at the focal point, and subsequent divergence of the optical beams into the far-field region (Fig. 3c and Supplementary Figs 7 and 8). The maximum number of rings observed with a certain angular range is then directly related to the focal length (f), the effective refractive index experienced by the SPPs ($n_{eff}$), and the wave-length (see Supplementary Note 6).

When the electron is placed 300 nm away from the center, the radiation is no longer normal to the structure but is slightly inclined (pos#1 at Fig. 5c). The inclination angle of the radiation is increased by further moving the electron beam away from the

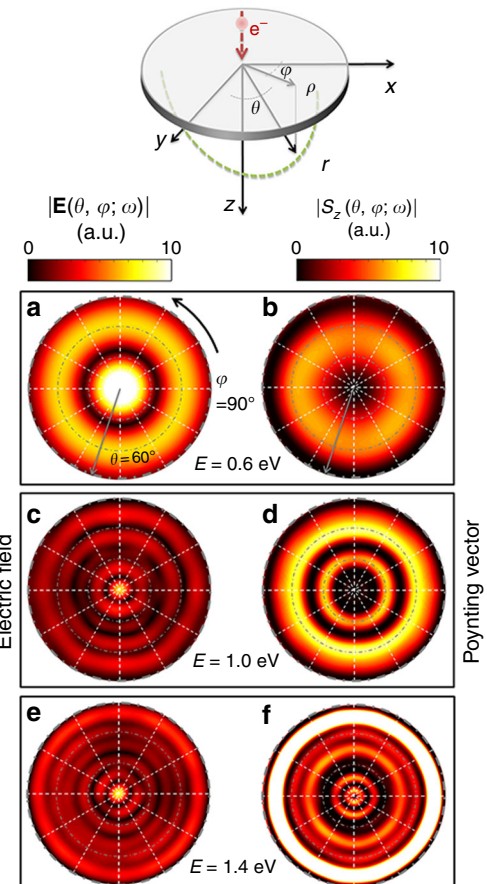

**Fig. 4** Simulated *k*-space characteristics of electron-induced radiation at the focal plane. **a**, **c**, **e** Amplitude of the electric field and **b**, **d**, **f** Poynting vector (corresponding to angular CL profile) at different energies

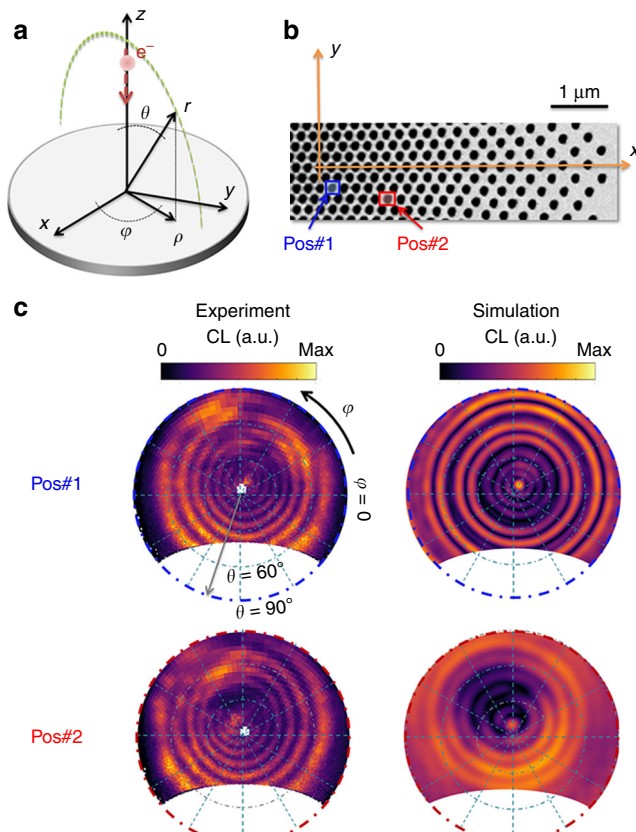

**Fig. 5** Far-field angle-resolved cathodoluminescence maps. **a** Detection geometry specified by the hemisphere positioned above the structure. **b** Impact positions for the electron beam, and **c** Experimental and simulated cathodoluminescence angular profile at *E* = 1.9 eV, at two different electron impact positions for the incident electron marked by red and blue colors in **b**, respectively

center of the structure; however, the radial symmetry of the radiation at the plane normal to the wave vector is almost preserved (pos#2 at Fig. 5c). This behavior is also confirmed by simulations, and gives excellent tunability of the polarization state and direction of emission of the (EDPHS) (Supplementary Note 4, Supplementary Figs 4 and 5). We have also investigated the effect of electron impact parameter for lenses with different focal lengths, which demonstrate the same sensitivity of the CL map to the electron impact position (see Supplementary Figs. 4 and 5). The measured CL intensity collected over the angular range in Fig. 5, integrated over a 0.12 eV energy range at *E* = 1.9 eV is $4.3 \times 10^{-5}$ photons/electron, which corresponds to $2.7 \times 10^5$ photons/s at a typical beam current of 1 nA Taking into account the calculated phase profile in the focal spot this corresponds to an intensity of 4.2 μW cm$^{-2}$dω at 1.9 eV. The calculated intensity at 1.9 eV is 6.5 μW cm$^{-2}$dω.

## Discussion
The diameter of the holes in our lens design is chosen to be much smaller than the wavelength of the light, so that diffraction from the holes is negligible, contrary to the case for Fresnel lenses and photon sieves[13] (for a comparison with photon sieves, see Supplementary Note 7 and Supplementary Figs. 9 to 11). This is confirmed by the fact that spectrally broadband behavior is observed. The lens thus operates by focusing the radiation as a result of the hyperbolic dispersion of the SPPs, which is due to the engineering of the spatial distribution of holes. Additionally, the contribution of SPPs to the focused radiation is most prominent

for the backward propagating SPPs reflected from the outer void rim, and not for the outgoing SPPs which are propagating directly after the excitations towards the rim (see the Supplementary Movie 1). We attribute this behavior to the inclination angle of the radiation from hyperbolic SPPs, which for the reflected SPPs from the void ring is directed towards the focal plane (See Supplementary Note 1 and Supplementary Fig. 1).

Finally, we compare the ultrafast EDPHS presented here with systems involving ultrafast laser excitation. An advantage of the EDPHS design in combination with spectral interferometry[35] is that the time between the electron excitation of the sample structure and the electron-generated optical probe is controlled on the attosecond scale. Furthermore, spectrally broadband excitation is possible, enabling interferometry from which time information can be derived. Alternatively, a pump-probe technique could be used, with the laser driving the electron cathode and the sample, as demonstrated by several groups[45,46]. However, in this case the electron pulse duration is typically on the order of sub-picoseconds so that the few-femtosecond dynamics of photonic nanostructures cannot be captured[47]. Fully laser-based designs have more flexibility to control intensity, narrow-band excitation wavelength, and polarization. But they miss the advantage of EDPHS that it is well suited in combination with electron microscopy. In the EDPHS case, the electron-induced sample excitation can be controlled at the nanoscale (both in the lateral dimensions *x*-*y* and along the symmetry axis *z*) and then correlated with the EDPHS excitation, which is not possible with

ultrafast laser spectroscopy. We note that our EDPHS design could be modified to create specially tailored states of light (linear, circularly polarized, angle-dependent). In fact, by controlling the beam position on the lens in the SEM such different states could potentially be made using a single geometry[34]. More precisely, even different structures can be fabricated on a single thin film for offering various states of light on demand, which can be further used for decomposition of photonic states of the sample.

In our EDPHS geometry time information will be derived from spectral interferometry, measuring the interference of radiation from the electron-excited sample and the broad continuum generated by the lens. In this way, a pulsed beam is not needed to derive time information; the interference is observed for every incoming electron. In a further advanced design it would be interesting to investigate if additional time control can be achieved by tailoring the time evolution of each individual electron wave packet. Structuring electron pulses by electromagnetic fields is an upcoming research field and may play a role in future EDPHS designs.

In summary, we have proposed and experimentally demonstrated a nanostructured planar lens which provides ultrafast focused electromagnetic radiation with a focal distance of only a few micrometers. The design principle offers large tunability in controlling both illumination angle and focal length. Due to the plasmon-induced mechanism of radiation, it acts as a coherent electron-driven photon source. In contrast to the diffractive metamaterial-based planar lenses, the proposed lens sustains an ultra-broadband spectral feature both in the near—and far—field and interestingly maintains its focusing ability for all spectral components. For this reason, we anticipate a wide range of applications for this structure, such as triggering an electron-induced mechanism of radiation in order to offer few-photon sources with a strong control on the collection efficiency and directionality of the emission, and on an ultrafast few fs time scale. Coherent control of the ultrafast responses of biological tissues can be facilitated by the concomitant electron–photon microscopy and spectroscopy in a single scanning electron microscope, as an example. Additionally, the proposed lens may be positioned directly in an electron microscope to facilitate electron-based spectral interferometry techniques[35]. Similarly, hybrid electron–photon pump-probe schemes become feasible in the future.

## Methods

**Fabrication of the electron-driven photon source**. First, a 50 nm thick gold film is deposited via electron beam evaporation on top of a 20 nm thick silicon nitride (Si$_3$N$_4$) TEM membrane. Due to the fragility of the membrane, structuring is performed by direct milling with a focused ion beam. In order to achieve the required high resolution we utilized the Raith ionLine Plus system as a dedicated high-resolution ion beam structuring tool. The holes where milled using a focused beam of doubly charged gold ions (Au + +) at an acceleration voltage of 35 kV and a beam-limiting aperture size of 7 μm. For some of the structures, the whole pattern is surrounded by a circular trench which is milled all the way through the gold film but only partially into the silicon nitride membrane itself.

**Time-dependent simulations**. For our purpose of realizing ultrashort electromagnetic sources, we have performed FDTD simulations, to gain an insight into the temporal distribution of the generated electron-induced radiation. The mesh size for FDTD simulation domain is 2 nm. We have modeled the electron probe with a time-dependent current source at the broadening of 1 nm[29,31]. The simulation domain has been terminated by using a higher order absorbing boundary conditions. Additionally, the dielectric function of gold has been modeled by a Drude function in addition to two critical point functions[48]. For calculating both CL and EELS maps, the field distributions along the electron trajectory and at a plane normal to that are calculated, respectively, and are then transformed into the frequency domain using discrete Fourier transformation. To achieve relaxation, 10,000 time iterations have been considered.

**EELS measurements**. We have used the Zeiss SESAM microscope for our low-loss EELS investigations. The electron probe has a kinetic energy of 200 keV and a size of a few nanometers. The acquisition time for each spectrum is 0.3 s. The microscope is equipped with an electron monochromator (CEOS Heidelberg) and the MANDOLINE energy filter. EELS data were acquired with an energy resolution of 70 meV as determined from the full width at half-maximum of the zero-loss peak (ZLP).

**CL measurements**. The CL setup is based on a scanning electron microscope with a Schottky field-emission gun (SFEG) electron source, equipped with an optical detection system. The kinetic energy of the electron beam is within the range of 1~30 keV. The primary light collection inside the microscope is performed by an off-axis aluminum paraboloid mirror that is mounted on a micromanipulation system. The mirror has an acceptance angle of 1.46π sr, a focal distance of 0.5 mm, and a 600 μm diameter hole above the focal point for supporting the electron excitation. For angle-resolved CL measurements the optical beam is passed through color filters to select a specific wavelength (bandwidth 40 nm), and directed onto a 2D back illuminated, Peltier-cooled, silicon CCD array[49]. As the holes were completely milled through the gold film and Si$_3$N$_4$ substrate, electrons pass through vacuum in the hole and therefore no incoherent radiation is expected from the substrate.

**Code availability**. The simulation toolbox used for this research is created in house and has been published under the granted European patent EP2784798B1.

## Data availability
The data sets within the article and Supplementary Information of the current study are available from the corresponding author upon reasonable request.

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

## Acknowledgements

M.H. and H.G. acknowledge financial support from the European Research Council (ERC Advanced Grant ComplexPlas), Bundesministerium für Bildung und Forschung, Deutsche Forschungsgemeinschaft (SPP1839), MWK Baden-Württemberg (IQST, ZAQuant), and Baden-Württemberg Stiftung. Work at AMOLF is part of the research program of the 'Nederlandse organisatie voor Wetenschappelijk Onderzoek' (NWO). It is also funded by the European Research Council (ERC). Work in Stuttgart Center for Electron Microscopy received funding from the European Union Horizon 2020 research and innovation programme under grant agreement No. 823717 – ESTEEM3. N.T. acknowledges financial support from the European Research Council (ERC Starting Grant NanoBeam).

## Author contributions

N.T. and H.G. discussed the initial concept. N.T. designed the lens and performed the simulations. S.G. acquired the EELS spectra, and S.M. did the CL measurements. M.H. fabricated the lens structure. N.T. wrote the manuscript with contributions from all the coauthors. A.P., H.G., and P.A.v.A. supervised the work and assisted in the interpretation of the results. All authors contributed to the final writing and the corrections of the manuscript.

## Additional information

**Competing interests:** A.P. is co-founder and co-owner of Delmic BV, a company that produces commercial cathodoluminescence systems like the one that was used in this work. The remaining authors declare no competing interests.

