## [Peer Review File · Nature Communications]

Reviewers' comments:

Reviewer #1 (Remarks to the Author):

Talebi et al. report a novel scheme to produce coherent broadband visible radiation from a planar metallic array of holes excited with swift relativistic electrons. The electrons passing the metallic structure lose energy, e.g. due to excitation of plasmonic modes in the gold metal, leading to the emission of light in a spectral range with photon energies from 0.4 to 1.2 eV (FWHM of the experimental results using EEL spectroscopy at 200 keV electron energy). The arrangement of the holes is optimized by a transformation optics approach such that it mimics the focusing properties of a hemispherical superlens structure, which was proposed by the first author in an earlier theoretical study [ref 30 in the manuscript].

The presented approach is innovative and smart, specifically in terms of the adaptation of the structure to sample dimensions which are well suited for electron microscopy, i.e. planar and thin. The experimental results are clear and show the potential of such electron-driven light sources. The simulations and calculations support the claims in the paper.

In principle, I am in favor to recommend this paper for publication in Nature Communications.

However, before I can come to a final decision, I have a few comments and questions.

I believe the proposed source could be potentially useful for electron microscopy, in particular in studies where external excitation of a specimen with broadband (ultrafast) laser radiation is not easy to manage.

However, in contrast to external excitation with short laser pulses, which are tunable in central wavelength, bandwidth, pulse duration, polarization, etc., and where the timing control of the pulses is possible, the presented approach seems to be quite inflexible in these respects. In addition, the ultrafast time structure claimed from the simulations can only be used (or proven) in experiments that involve a pulsed (ultrafast) electron source. The authors should discuss the possibilities to control the radiation and make use of the time structure for time-resolved experiments.

Another important aspect for the usability of this type of source is the intensity in the focal spot. The authors should estimate this quantity from the measured far-field intensity of the CL signal. It should be possible to reconstruct the focal intensity profile from the measured CL far-field intensity by taking into account the emission phase at the sample. If I understood correctly, this phase has been calculated already for the simulation in Figure 4.

The stated spectral bandwidth in the abstract (0.4- 2.6 eV) refers to the full range (not FWHM) of the calculated spectrum in Figure 1, for the hemispherical structure excited with 30 keV electron energy. However, the experimental EEL spectra are obtained with the planar structure and 200-keV electron energy. Whereas the CL signal measured at 1.9 eV seems to indicate an even broader spectrum, the authors should state in the abstract at least the FWHM bandwidth of calculated spectrum, if not just the measured EELS spectrum, which is centered at 0.6 eV and has a width of 0.8 eV.

The presentation of the EEL spectra in Figure 2c should be merged into one graph, to allow for a better comparability of the experimental results with the simulations.

The experimental spectra in Figure 2c seem to be broader than the simulated ones. Is this due to incoherent processes?

More generally, incoherent CL (e.g. from the substrate) should be a strong contribution in such experiments. Did the authors analyze the background signal in the CL measurements to estimate the incoherent signal contribution?

Some minor technical comments:

The numbering/order of the figures in the supplement is not correct. For example, the figures S2 and S3 referenced in the main text should be S3 and S4 instead, respectively.

In the Methods, under the point CL measurements, it is stated: "... a focal distance of 0.5 mm, and a 600 mm diameter hole ... ". Are these numbers correct? This seems to be a weird aspect ratio for an off-axis parabolic mirror.

Reviewer #2 (Remarks to the Author):

The authors present here a plasmonic structure arranged of a nanohole array in a gold film that generates ultrashort chirped electromagnetic wave packets upon 30-200 keV electron irradiation. The array is designed using transformation optics; the approach is suitable for designing other lenses in modeling frameworks that can be coupled to exact full-wave EM simulators in the time-domain [the Finite Difference Time Domain (FDTD) method in particular]. While this multiphysics approach has been adopted by the authors before (e.g., Ref. [1] of this Review, and references within) to my knowledge suitable models in such multiphysics frameworks for relativistic electron beams producing optical radiation, when interacting with tailored nanostructures are missing. Such a multiphysics approach that is experimentally validated here allows capturing the behavior of electron excitation interfaced with nanopatterned plasmonic metasurface, which is not attainable otherwise. Indeed, the predictive potential of the multiphysics framework developed earlier culminated in impressive experiment-to-model match depicted in Figs. 2c and 5c.

If developed right this work will enable transformative possibilities in designing nano-structured material platforms for coherent electron-driven photon sources that are relevant to a range of applications. It could also provide invaluable insights into the dynamic behavior of such hybrid devices that complement optical experiments.

Based on the above we find that present work is of broad and high interest of the NC community and thus suitable for the NC audience.

However, some parts of the paper are somewhat inaccessible for a non-specialist; this is particularly important for Introduction which should be reaching out to larger audiences. We also have some questions concerning the technical implementation details; we would think that providing more specific details would benefit the community and the broader application of this innovative framework validated here.

COMMENTS:

(1) Introduction I. The authors may want to include at least a bit more detailed introduction to the effect they are employing. They may have a special affinity to the papers they cite on the core effects [21, 22, 23], but a good practice is to cite first the original papers on the effect in hand, e.g., start with Phys. Rev. 92 1069 (1953), and then cite [23]. Generally, assuming that the self-citation level in this manuscript is outrageous, it would be perhaps fair to give some room to earlier classical papers.

(2) Introduction II. The authors shall spend enough time comparing their effort with the state-of-the-art studies that they mention on the fly with Ref. [18-20]. The introductory part should not become a review article, but the audience (without visiting the library) needs to know what is new and how the authors are advancing the field. The brief and lonely phrase, "We avoid diffractive metamaterial elements [18, 19] in order to achieve ultrafast control and unidirectional emission and exploit hyperbolic dispersion to create efficient plasmon radiation," in our opinion explains nothing and is structurally confusing. The direct, to the point comparison vs. [18, 19] would be of importance.

(3) Transformation Optics. This part is explained quite lightly. We expected to get some more details from the supplement but did not get anything. It would be instrumental to get more details on the TO mapping and the associated geometry of the lens in Supplement. It also seems that there could be some resemblance to a standard mapping projection. Another critical question to address is, do we gain something through using TO and vs. traditional metasurface design alternative?

(4) Time-dependent simulations. The authors write, "Additionally, the dielectric function of gold has been modeled by a Drude function in addition to two critical point functions [40]." First, the sentence should cite Ref. [39] (P. G. Etchegoin, E. C. Le Ru, M. Meyer, An analytic model for the optical properties of gold, *The Journal of Chemical Physics* 125, 164705 (2006).) Moreover, Ref. [39] should be complemented by its Erratum, published in *The Journal of Chemical Physics* 127, 189901 (2007). The authors should be advised that by mistake the parameters of Table I in [39] are obtained from a slightly different equation to Eq. 3 of Ref. [39]. These incorrect parameters of [39] result in a reasonable representation of the optical properties of gold, but not as accurate as a fit in Fig. 1 of [39] would indicate. Correct Table I is reproduced in Erratum.

[1] N. Talebi, Electron-light interactions beyond the adiabatic approximation: recoil engineering and spectral interferometry, *Advances in Physics: X*, 3 (2018), p. 1499438.

Reviewer 1:

Talebi et al. report a novel scheme to produce coherent broadband visible radiation from a planar metallic array of holes excited with swift relativistic electrons. The electrons passing the metallic structure loose energy, e.g due to excitation of plasmonic modes in the gold metal, leading to the emission of light in a spectral range with photon energies from 0.4 to 1.2 eV (FWHM of the experimental results using EEL spectroscopy at 200 keV electron energy). The arrangement of the holes is optimized by a transformation optics approach such that it mimics the focusing properties of a hemispherical superlens structure, which was proposed by the first author in an earlier theoretical study [ref 30 in the manuscript].

The presented approach is innovative and smart, specifically in terms of the adaptation of the structure to sample dimensions which are well suited for electron microscopy, i.e. planar and thin. The experimental results are clear and show the potential of such electron-driven light sources. The simulations and calculations support the claims in the paper.

In principle, I am in favor to recommend this paper for publication in Nature Communications. However, before I can come to a final decision, I have a few comments and questions.

Our response:

We gratefully thank Reviewer 1 for the constructive feedback and the positive assessment of our work. We also believe that the comments raised by the Reviewer have helped us to improve our manuscript.

Comment 1:

"I believe the proposed source could be potentially useful for electron microscopy, in particular in studies where external excitation of a specimen with broadband (ultrafast) laser radiation is not easy to manage. However, in contrast to external excitation with short laser pulses, which are tunable in central wavelength, bandwidth, pulse duration, polarization, etc., and where the timing control of the pulses is possible, the presented approach seems to be quite inflexible in these respects."

Our response:

Thank you very much for these detailed remarks. There are clear conceptual differences - and advantages and disadvantages - between the ultrafast electron-driven photon source presented here and geometries involving ultrafast laser excitation. Advantages of the EDPHS geometry in combination with spectral interferometry are the possibilities to investigate correlations and improve coherence and therefore temporal resolution. It was not the purpose of this manuscript to deliver a one-by-one comparison between the laser-induced electron spectroscopy techniques and our approach. However, thanks to this comment, we found the opportunity to provide a much more detailed outlook.

Actions taken:

We have added this paragraph to the text before the Conclusion to highlight the main points:

"Finally, we compare the ultrafast EDPHS presented here with systems involving ultrafast laser excitation. An advantage of the EDPHS design in combination with spectral interferometry¹ is that the time between the electron excitation of the sample structure and the electron-generated optical probe is controlled on the attosecond scale. Furthermore, spectrally broadband excitation is possible, enabling interferometry from which time information can be derived. Alternatively, a pump-probe technique could be used, with the laser driving the electron cathode and the sample, as demonstrated by several groups^{2,3}. However, in this case the electron pulse duration is typically on the order of sub-picoseconds so that the few-femtosecond dynamics of photonic nanostructures cannot be captured⁴. Fully laser-based designs have more flexibility to control intensity, narrow-band excitation wavelength, and polarization. But they miss the advantage of EDPHS that it is well suited in combination with electron microscopy. In the EDPHS case, the electron-induced sample excitation can be controlled at the nanoscale (both in the lateral dimensions x-y and along the symmetry axis z) and then correlated with the EDPHS excitation, which is not possible with ultrafast laser spectroscopy. We note that our EDPHS design could be modified to create specially tailored states of light (linear, circularly polarized, angle-dependent). In fact, by controlling the beam position on the lens in the SEM such different states could potentially be made using a single geometry.⁵ More precisely, even different structures can be fabricated on a single thin film for offering various states of light on demand, which can be further used for decomposition of photonic states of the sample."

Comment 2:

"In addition, the ultrafast time structure claimed from the simulations can only be used (or proven) in experiments that involve a pulsed (ultrafast) electron source. The authors should discuss the possibilities to control the radiation and make use of the time structure for time-resolved experiments."

Our response:

The interference between the CL emissions of the sample and the EDPHS is used to prove the time structure of the radiation here. Furthermore this time structure is perfectly controlled by the interaction time, and the design of the lens and not by the time structure of the electron pulses. This has been specifically described in the main text as below.

Actions taken:

We added this paragraph to the text before the Conclusion:

"In our EDPHS geometry time information will be derived from spectral interferometry, measuring the interference of radiation from the electron-excited sample and the broad continuum generated by the lens. In this way, a pulsed beam is not needed to derive time information; the interference is observed for every incoming electron. In a further advanced design it would be interesting to investigate if additional time control can be achieved by tailoring the time evolution of each individual electron wave packet. Structuring electron pulses by electromagnetic fields is an upcoming research field and may play a role in future EDPHS designs."

Comment 3:

"Another important aspect for the usability of this type of source is the intensity in the focal spot. The authors should estimate this quantity from the measured far-field intensity of the CL signal. It should be possible to reconstruct the focal intensity profile from the measured CL far-field intensity by taking into account the emission phase at the sample. If I understood correctly, this phase has been calculated already for the simulation in Figure 4."

Our response:

We specifically thank the reviewer for this important remark. The measured photon flux at the CL detector is 4.3×10^{-5} photons/electron at $E=1.9$ eV, integrated over 0.12 eV spectral bandwidth. This amounts to a focal intensity of $4.2 \mu\text{W cm}^{-2} d\omega$ which is in quite good agreement with the calculated intensity of $6.5 \mu\text{W cm}^{-2} d\omega$.

Actions taken:

We added the following paragraph to the main text:

"The CL intensity collected over the angular range in Fig. 5, integrated over a 0.12 eV energy range at $E = 1.9$ eV, is 4.3×10^{-5} photons/electron which corresponds to 2.7×10^5 photons/s at a typical beam current of 1 nA. Taking into account the calculated phase profile in the focal spot this corresponds to an intensity of $4.2 \mu\text{W cm}^{-2} d\omega$ at 1.9 eV, per electron excitation. The calculated intensity at 1.9 eV is $6.5 \mu\text{W cm}^{-2} d\omega$."

Comment 4:

"The stated spectral bandwidth in the abstract (0.4- 2.6 eV) refers to the full range (not FWHM) of the calculated spectrum in Figure 1, for the hemispherical structure excited with 30 keV electron energy. However, the experimental EEL spectra are obtained with the planar structure and 200-keV electron energy. Whereas the CL signal measured at 1.9 eV seems to indicate an even broader spectrum, the authors should state in the abstract at least the FWHM bandwidth of calculated spectrum, if not just the measured EELS spectrum, which is centered at 0.6 eV and has a width of 0.8 eV."

Our response:

We thank Reviewer 1 for mentioning this inconsistency.

Actions taken:

The inconsistency was corrected as mentioned by the reviewer by considering the experimental EELS spectra as follows. We added the bandwidth of 0.4-2.6 eV to the abstract by changing the sentence

“They decay by radiation in a spectral band covering a bandwidth from 0.4-2.6 eV which is focused into a 1.5 micrometer beam waist.”

to

“They decay by radiation in a spectral band centered around 0.6 eV covering a bandwidth of 0.8 eV which is focused into a 1.5 micrometer beam waist.”

Comment 5:

“The presentation of the EEL spectra in Figure 2c should be merged into one graph, to allow for a better comparability of the experimental results with the simulations.

The experimental spectra in Figure 2c seem to be broader than the simulated ones. Is this due to incoherent processes?”

Our response:

We merged the experimental and simulation results as requested by the reviewer. This enables a better comparison between simulations and experimental results. The broadening of the experimental data is indeed different to that of numerical spectra, and it is related to the well-known instrumental zero-loss peak, as well as scattering of the polaritons by the surface roughness.

Actions taken:

The last paragraph in the section EELS Experiment, has been extended with

“However, experimental EELS data show slightly broader resonances, which are due to the convolution of the zero-loss peak with plasmon resonances⁶. This zero-loss peak has contribution from electrons that did not undergo inelastic scattering, but may have been scattered elastically or with an energy loss too small to measure. Moreover, surface roughness in the sputtered gold thin film increases the damping of surface plasmons and hence results in an additional broadening of the EELS spectra.”

Comment 6:

“More generally, incoherent CL (e.g., from the substrate) should be a strong contribution in such experiments. Did the authors analyze the background signal in the CL measurements to estimate the incoherent signal contribution?”

Our response:

The electron lens was fabricated by focused-ion beam milling of holes in an Au film deposited on a 20 nm thick Si₃N₄ membrane. The holes were completely milled through, so that the electron passes through vacuum in the hole. Therefore no incoherent radiation is expected.

Actions taken:

We added this short description to the text, at the method section:

“As the holes were completely milled through the gold film and the Si₃N₄ substrate, electrons pass through vacuum in the hole and therefore no incoherent radiation is expected from the substrate.”

Comment 7:

“Some minor technical comments:

The numbering/order of the figures in the supplement is not correct. For example, the figures S2 and S3 referenced in the main text should be S3 and S4 instead, respectively.”

Our Response:

We gratefully thank the reviewer for these careful remarks. We corrected the misleading citations to the supplementary figures in many places throughout the manuscript.

Actions Taken:

The figure captions and references to the supplementary figures have been corrected in different places.

Comment 8:

"In the Methods, under the point CL measurements, it is stated: "... a focal distance of 0.5 mm, and a 600 mm diameter hole ... ". Are these numbers correct? This seems to be a weird aspect ratio for an off-axis parabolic mirror."

Our Response:

We also thank the reviewer for pointing out this typo. We corrected the dimension of the hole in the mirror.

Actions Taken:

The sentence

"a focal distance of 0.5 mm, and a 600 mm diameter hole above the focal point for supporting the electron excitation."

Has been corrected to

"a focal distance of 0.5 mm, and a 600 μm diameter hole above the focal point for supporting the electron excitation."

Reviewer 2:

The authors present here a plasmonic structure arranged of a nanohole array in a gold film that generates ultrashort chirped electromagnetic wave packets upon 30-200 keV electron irradiation. The array is designed using transformation optics; the approach is suitable for designing other lenses in modeling frameworks that can be coupled to exact full-wave EM simulators in the time-domain [the Finite Difference Time Domain (FDTD) method in particular]. While this multiphysics approach has been adopted by the authors before (e.g., Ref. [1] of this Review, and references within) to my knowledge suitable models in such multiphysics frameworks for relativistic electron beams producing optical radiation, when interacting with tailored nanostructures are missing. Such a multiphysics approach that is experimentally validated here allows capturing the behavior of electron excitation interfaced with nanopatterned plasmonic metasurface, which is not attainable otherwise. Indeed, the predictive potential of the multiphysics framework developed earlier culminated in impressive experiment-to-model match depicted in Figs. 2c and 5c.

If developed right this work will enable transformative possibilities in designing nano-structured material platforms for coherent electron-driven photon sources that are relevant to a range of applications. It could also provide invaluable insights into the dynamic behavior of such hybrid devices that complement optical experiments.

Based on the above we find that present work is of broad and high interest of the NC community and thus suitable for the NC audience.

However, some parts of the paper are somewhat inaccessible for a non-specialist; this is particularly important for Introduction which should be reaching out to larger audiences. We also have some questions concerning the technical implementation details; we would think that providing more specific details would benefit the community and the broader application of this innovative framework validated here.

Our response:

We gratefully thank the reviewer for the kind and positive response. We also appreciate his/her careful assessment of the manuscript and proposing ways to enhance the broad readership of our manuscript.

Comment 1:

“Introduction I. The authors may want to include at least a bit more detailed introduction to the effect they are employing. They may have a special affinity to the papers they cite on the core effects [21, 22, 23], but a good practice is to cite first the original papers on the effect in hand, e.g., start with Phys. Rev. 92 1069 (1953), and then cite [23]. Generally, assuming that the self-citation level in this manuscript is outrageous, it would be perhaps fair to give some room to earlier classical papers.”

Our response:

Thank you very much for these important suggestions. We have now added several citations to the manuscript, including also classical papers.

Actions taken (see also reviewer 1, comment 2):**The sentence**

“Recently, diffractive metamaterial lenses have been applied to control the directionality and the polarization states of electron-induced radiation.”

has been moved to the end of the paragraph, to first describe mechanisms of radiation in more detail and to cite additional relevant papers. Additionally, we added the following paragraph to the introduction to describe in more detail the generation of radiation by electrons in our geometry:

“Transition radiation occurs when an electron traverses a metallic surface, due to the sudden annihilation of the induced dipole formed by the electron and its image charge. In addition to transition radiation, electrons launch plasmon polaritons along the surface of a metal. These induced polaritons can couple to radiation by scattering from gratings and defects.”

The following citations were also added to the introduction at an appropriate order:

- [1] Larmor, J. LXIII. On the theory of the magnetic influence on spectra; and on the radiation from moving ions. *The London, Edinburgh, and Dublin Philosophical Magazine and Journal of Science* **44**, 503-512, doi:10.1080/14786449708621095 (1897).
- [2] Ginzburg, V. L. in *Progress in Optics* Vol. 32 (ed E. Wolf) 267-312 (Elsevier, 1993).
- [3] Smith, S. J. & Purcell, E. M. Visible Light from Localized Surface Charges Moving across a Grating. *Phys Rev* **92**, 1069-1069, doi:DOI 10.1103/PhysRev.92.1069 (1953).
- [4] Bachheimer, J. P. Experimental Investigation of the Interaction Radiation of a Moving Electron with a Metallic Grating: The Smith-Purcell Effect. *Phys Rev B* **6**, 2985-2994, doi:10.1103/PhysRevB.6.2985 (1972).
- [5] Salisbury, W. W. Generation of Light from Free Electrons*. *J. Opt. Soc. Am.* **60**, 1279-1284, doi:10.1364/JOSA.60.001279 (1970).

Comment 2:

"Introduction II. The authors shall spend enough time comparing their effort with the state-of-the-art studies that they mention on the fly with Ref. [18-20]. The introductory part should not become a review article, but the audience (without visiting the library) needs to know what is new and how the authors are advancing the field. The brief and lonely phrase, "We avoid diffractive metamaterial elements [18, 19] in order to achieve ultrafast control and unidirectional emission and exploit hyperbolic dispersion to create efficient plasmon radiation," in our opinion explains nothing and is structurally confusing. The direct, to the point comparison vs. [18, 19] would be of importance."

Our response:

Thank you for pointing out this lack of clarity in our wording. We agree that the sketch in Figure 1 by itself is not providing sufficient detail of our design concept. We have expanded the text to better describe the differences between our approach and other design principles.

Actions taken:

The mentioned paragraph in the introduction part has been revised to:

"Two-dimensional arrays of split-ring resonators were utilized earlier for enhancing the radiation at a rather wide spectral range (0.6 eV bandwidth) due to the broad plasmonic resonances in gold. However, the emission pattern of EDPHS based on these resonators sustains a wide angular range. In contrast, holographic designs are perfectly suited for controlling the directionality of the emission. In this approach, the interference pattern of the electron-induced plasmons at the gold/air interface with a light field at a specific wavelength and with a desired shape is used to generate the required hologram; however this method is highly frequency-selective."

We also modified this sentence

"Our design principle is rather based on the simple geometrical manifestation of the principles of transformation optics."^{7 6 4 2}

Into

"Our design principle is rather based on the geometrical manifestation of the principle of transformation optics, by manipulating the effective refractive index of a gold thin film using incorporated holes with dimensions much smaller than the wavelength of the emitted light pulse. Additionally, the realized system mimics an anisotropic thin film, with different signs for in-plane and normal permittivity components, thus the structure is effectively a hyperbolic material. Consequently, a branch of polaritons propagating along the surface of our engineered thin film is mapped into the light cone which largely enhances the electron-induced radiation."

Comment 3:

"Transformation Optics. This part is explained quite lightly. We expected to get some more details from the supplement but did not get anything. It would be instrumental to get more details on the TO mapping and the associated geometry of the lens in Supplement. It also seems that there could be some resemblance to a

standard mapping projection. Another critical question to address is, do we gain something through using TO and vs. traditional metasurface design alternative?"

Our response:

Thank you for mentioning this problem. Indeed there are quite some similarities between projection algorithm and transformation optics, as we have projected a hexagonal lattice of holes from a spherical coordinate system to the cylindrical coordinate system, to effectively engineer the refractive index of our thin film.

Actions taken:

We addressed this comment by adding a new section 7 in the supplementary section, where we concisely described the basics of design principles based on a diffractive photon sieve analysis and compared that to our TO mapping algorithm. We repeat this chapter at the end of response letter for convenience in the review process (see Appendix 1):

Comment 4:

"Time-dependent simulations. The authors write, "Additionally, the dielectric function of gold has been modeled by a Drude function in addition to two critical point functions [40]." First, the sentence should cite Ref. [39] (P. G. Etchegoin, E. C. Le Ru, M. Meyer, An analytic model for the optical properties of gold, The Journal of Chemical Physics 125, 164705 (2006).) Moreover, Ref. [39] should be complemented by its Erratum, published in The Journal of Chemical Physics 127, 189901 (2007). The authors should be advised that by mistake the parameters of Table I in [39] are obtained from a slightly different equation to Eq. 3 of Ref. [39]. These incorrect parameters of [39] result in a reasonable representation of the optical properties of gold, but not as accurate as a fit in Fig. 1 of [39] would indicate. Correct Table I is reproduced in Erratum."

Our response:

Thank you for mentioning the typo in citing the correct reference. Indeed the correct model for gold has been used by us.

Actions taken:

We cited the correct paper by Etchegoin accompanied by its Erratum.

References:

- 1 Talebi, N. Spectral Interferometry with Electron Microscopes. *Sci Rep-Uk* **6**, doi:ARTN 33874 10.1038/srep33874 (2016).
- 2 da Silva, N. R. *et al.* Nanoscale Mapping of Ultrafast Magnetization Dynamics with Femtosecond Lorentz Microscopy. *Phys Rev X* **8**, doi:Artn 031052 10.1103/Physrevx.8.031052 (2018).
- 3 Vanacore, G. M. *et al.* Attosecond coherent control of free-electron wave functions using semi-infinite light fields. *Nat Commun* **9**, doi:Artn 2694 10.1038/S41467-018-05021-X (2018).
- 4 Feist, A. *et al.* Ultrafast transmission electron microscopy using a laser-driven field emitter: Femtosecond resolution with a high coherence electron beam. *Ultramicroscopy* **176**, 63-73, doi:https://doi.org/10.1016/j.ultramic.2016.12.005 (2017).
- 5 Osorio, C. I., Coenen, T., Brenny, B. J. M., Polman, A. & Koenderink, A. F. Angle-Resolved Cathodoluminescence Imaging Polarimetry. *Acs Photonics* **3**, 147-154, doi:10.1021/acsp Photonics.5b00596 (2016).
- 6 Konečná, A. *et al.* Vibrational electron energy loss spectroscopy in truncated dielectric slabs. *Phys Rev B* **98**, 205409, doi:10.1103/PhysRevB.98.205409 (2018).
- 7 Kundtz, N. & Smith, D. R. Extreme-angle broadband metamaterial lens. *Nat Mater* **9**, 129-132, doi:10.1038/Nmat2610 (2010).

Appendix 1:

Transformation Optics and Differences with photon sieves

We first discuss the possibilities for designing an EDPHS which operates by the diffraction of propagating plasmons from incorporated holes. There are a few similarities between the structure discussed here and photon sieves. The latter are based on the diffraction of free-space waves by embedded holes in dielectric thin films, and have been established as an efficient way for focusing x-rays (Fig. S8 (a))⁸. In the notation of Fig. S8 (a), radiation from a point source S is focused at point P , by means of a structured thin film consisting of a distribution of holes. The size of these transmissive pinholes and their distribution are both chosen in such a way to allow for constructive interference of the diffracted rays at the focal point P . To allow for this, the distribution of the pinholes should satisfy $\sqrt{r_n^2 + S^2} + \sqrt{r_n^2 + f^2} = S + f + n\lambda$. Here r_n is the distance of the n^{th} hole from the origin, λ is the wavelength of the source, and n is an integer. Obviously, the focusing ability strongly depends on the wavelength of the source which hinders photon sieves from offering a broadband response.

In a similar sieve geometry for a moving electron interacting with a metallic thin film, we consider excitation of plasmon polaritons at the impact position of the electron (Fig. S8 (b)). A distribution of pinholes (or other diffractive centers like metamaterial elements or ribs) is embedded to allow for outcoupling of the propagating plasmons, creating a radiation continuum. For constructive interference of the outcoupled beams at the focal plane, the distribution of the diffractive elements (here pinholes) should satisfy

$$n_{\text{eff}} r_n + \sqrt{r_n^2 + f^2} = f + n\lambda \quad (\text{S.6})$$

where n_{eff} is the effective mode index of the plasmons in the structured thin film. It should be noted that for both cases (photon sieves for either x rays or plasmons), the size of the pinholes should be large enough to allow for either transmissive response or an efficient outcoupling of the plasmon polaritons; i.e., the diameter of the holes should be comparable to the wavelength of the source or excited plasmons.

Fig. S8. Schematic of the focusing capability of a photon sieve for (a) x-rays, and (b) electron-induced plasmon polaritons.

In the following we show how the photon sieve design principle can be used to focus the electron-induced radiation. We consider here a 80 nm-thick Al film and use eq. (S.6) to calculate the required distribution of the pinholes to facilitate focusing at $\lambda = 206.6 \text{ nm}$ ($E = 6 \text{ eV}$). At this wavelength, the 80-nm Al film sustains plasmon polaritons with a mode index of $n_{\text{eff}} = 1.106 + i0.015$ (for this thickness, plasmon polaritons at the upper and lower surfaces are uncoupled and the even and odd modes are degenerate). Furthermore, the diameter of the pinholes is varied between $D = 60 \text{ nm}$ and $D = 120 \text{ nm}$ (Fig. S9 (a)). The radiation from a non-structured Al thin film interacting with a moving electron covers a broad angular range (Fig. S9 (b)), though

by imbedding the pinholes with the desired distribution, we achieve focusing at $f = 900\text{nm}$ away from the structured film (Fig. S9 (c)). However, at a slightly different energy of $E = 6.3\text{eV}$, the radiation becomes completely unfocused (Fig. S9 (d)).

Fig. S9. (a) Distribution of holes to achieve focused radiation at $f = 900\text{nm}$ and $E = 6\text{eV}$, for an Al thin film. The z-component of the electric field at a given time for (b) a non-structured and (c) structured Al film at $E = 6\text{eV}$, and (d) for a structured Al film at $E = 6.3\text{eV}$.

Given the unwanted sensitivity of the photon sieves to the wavelength, we exploit other possibilities which avoid such sensitivity from the design principle. As discussed in the main text, focusing capabilities based on geometrical refraction, like focusing by a hemispherical thin film, are inherently broadband in nature. We therefore here exploit principles of transformation optics to generate a distribution of non-transmissive pinholes, with the aim to engineer the effective mode index of plasmons polaritons to mimic the response of a hemisphere. The general assumption is based on adiabatically tuning the periodicity of the lattice of holes, to achieve an engineered mode index as a function of distance from the origin. Using the Jacobian of the transformation matrix Λ for mapping a function from a spherical coordinate system with coordinates (r, θ, φ) to a cylindrical system with coordinates (ρ, φ, z) (with φ and θ the zenithal and azimuthal angles, respectively), the permittivity of the structure in the cylindrical system $\epsilon' = |\Lambda|^{-1} \Lambda \hat{\epsilon} \Lambda^T$ is derived from the permittivity of the material in spherical coordinates ($\hat{\epsilon} = \epsilon_{\text{Au}} \mathbf{I}$, where ϵ_{Au} is the permittivity of gold and \mathbf{I} is the identity matrix). Specifically, we take $\epsilon'_{\rho\rho} = \epsilon_{\text{Au}} \rho^2 / (\rho^2 + d^2)$, $\epsilon'_{\rho\varphi} = \epsilon'_{zz} = 0$, and $\epsilon'_{\varphi\varphi} = \epsilon_{\text{Au}}$. Formally, such a structure seems practically out of reach, considering that $\epsilon'_{zz} = 0$. Nevertheless $\epsilon'_{\rho\rho}$ dominates the propagation of polaritons and can be suitably engineered. For a hemisphere with the focal point at the center, we set for our mapping purpose $d = R$, so that $\epsilon'_{\rho\rho} = \epsilon_{\text{Au}} \frac{(\rho/R)^2}{(\rho/R)^2 + 1} = \epsilon_{\text{Au}} \sin^2 \theta$. At

$\rho \gg f$, $\epsilon'_{\rho\rho} \approx \epsilon_{\text{Au}}$. In other words the distance between the holes should be adiabatically increased for increasing distance from the origin, to obtain the pure gold permittivity at $\rho \gg f$. A geometric projection algorithm which maps the position of the holes on the gold hemispherical film onto a thin film located at

$z = d = R$ is used to obtain such a distribution of pinholes (see Fig. S10 (a)). We assume a hexagonal lattice of holes with centers located at (x_i, y_i) as shown in Fig. S10 (b) and use the mapping algorithm to obtain the new

Fig. S10. (a) Mapping principle for projecting the lattice of pinholes from a spherical domain to a planar thin film. The 2-dimensional distribution of the lattice points at the (b) original coordination and (b) final coordination. (c) The final distribution of lattice points for the fabricated EDPHS.

distribution for the holes in the thin film as $\rho'_i = R \tan \theta_i$ and $\tan \theta_i = \rho_i / \sqrt{R^2 - \rho_i^2}$ (Fig. S10 (c)). To maintain an azimuthally symmetric radiation pattern only the projected holes within a certain radius are included in the design. For $R \rightarrow \infty$ the original lattice will be unambiguously maintained in the projected domain as well (a hemisphere film with $R \rightarrow \infty$ corresponds to a flat film). Finally, for the EDPHS investigated here, we assumed $R = 5.7 \mu\text{m}$, for which the lattice in Fig. S10 (d) is obtained. At each lattice point a pinhole with a diameter of 100 nm is considered. The overall lattice is incorporated onto an Au thin film with a thickness of 50 nm.

REVIEWERS' COMMENTS:

Reviewer #1 (Remarks to the Author):

The authors have sufficiently revised the manuscript according to both Reviewers comments and answered all open questions.

Therefore, I recommend the current version of the paper for publication in Nature Communications.

Reviewer #2 (Remarks to the Author):

We're happy with the authors' response. The paper has been vastly improved. Please publish.

Nature Communications Manuscript NCOMMS-18-26401
Response to the Reviewers

Reviewer 1

The authors have sufficiently revised the manuscript according to both Reviewers comments and answered all open questions.

Therefore, I recommend the current version of the paper for publication in Nature Communications.

We gratefully thank the reviewer for the positive assessments of our work and recommendation for publications!

Reviewer #2 (Remarks to the Author):

We're happy with the authors' response. The paper has been vastly improved. Please publish.

Many thanks fort the supportive and positive comments and also recommendation of our manuscript for publication in Nature Communications!